# Self-Optimizing Traffic Steering for 5G mmWave Heterogeneous Networks

**DOI:** 10.3390/s22197112

**Published:** 2022-09-20

**Authors:** Jun Zeng, Hao Wang, Wei Luo

**Affiliations:** 1The School of Electrical Information, Wuhan University, Wuhan 430072, China; 2Fiberhome Communication Technology Co., Ltd., Wuhan 430074, China; 3Wuhan Second Ship Design and Research Institute, Wuhan 430064, China; 4China Ship Development and Design Center, Wuhan 430064, China

**Keywords:** 5G heterogeneous network, mmWave communication, traffic steering, self-optimization

## Abstract

Driven by growing mobile traffic, millimeter wave (mmWave) communications have recently been developed to enhance wireless network capacity. Due to insufficient coverage and the lack of support for mobility, mmWave is often deployed in the ultra-dense small cells of the 5G heterogeneous network. In this article, we first summarize the characteristics of the 5G heterogeneous network from the viewpoints of devices, spectra, and networks. We then propose a triple-band network structure which incorporates licensed bands, sub-6GHz unlicensed bands, and mmWave bands to support various types of mobile users. Based on the novel network structure, we further propose a self-optimizing traffic steering strategy which can intelligently steer traffic to specific networks and spectra according to the dynamic network and traffic environments. Several use cases are also discussed to facilitate the implementation of our proposals. Finally, we present numerical results to demonstrate that the proposed network structure and strategy can effectively enhance the system throughput and energy efficiency.

## 1. Introduction

Recently, the exponential growth of mobile data traffic has culminated in the development of fifth-generation (5G) wireless communication systems, with the ultimate goals of gigabit-level data rate, millisecond latency, billions of connected devices, and so on. Confronting these demands, academia and industry have developed several technologies, such as massive multiple-input multiple-output (MIMO), full-duplex (FD) communication, satellite communication [1,2], and network densification. However, these cutting-edge techniques still cannot meet the ever-increasing demands for wireless traffic, which is expected to increase by about 1000 fold by the year 2020. Meanwhile, the current available wireless spectrum is almost saturated even if cognitive radio technology is applied, leaving limited room to exploit. Hence, it is imperative to provide a paradigm shift from the wireless spectrum to higher frequencies. With this regard, the millimeter wave (mmWave) band with abundant unused or lightly used bandwidths becomes a viable way to satisfy 5G communication demands [3].

The availability of rich spectrum resources makes mmWave communications a predominant technique to acquire gigabit-level data rates, millisecond low latency, and billions of connected devices in the 5G era. Moreover, owing to the short wavelength, mmWave communicatiosn can realize directional pencil-beam transmission with limited interference [4,5]. However, the practical implementation of mmWave communications still faces various challenges, such as high propagation loss, susceptibility to blockage in daily environments (such as buildings, walls, humans, foliage, and even rain drops), and high hardware complexity [6,7]. These challenges historically make utilizing mmWave communications infeasible for mobile communications due to insufficient coverage and a lack of support for mobility, especially in non-line-of-sight (NLOS) environments [8]. However, mmWave communications are much more applicable for the 5G heterogeneous network (HetNet) with ultra-dense small cell deployment [9,10].

On the other hand, with the ultra-dense deployment of mmWave access points (APs), self-optimizing network management mechanisms are essential to improve system performance while simultaneously reducing both operational expenditure and capital expenditure [11,12]. Therefore, in this article, we specifically investigate self-optimizing traffic steering in an mmWave HetNet. The goal of traffic steering is to optimally and intelligently allocate traffic to different networks and different spectra to achieve diverse optimization objectives [13]. There have been several existing works studying the application of traffic steering in different scenarios, such as LTE in unlicensed networks (LTE-U) [14] and LTE HetNets [15]. However, to the best of our knowledge, there still has been no work that investigates traffic steering in the mmWave-based 5G HetNet.

Different from the traditional dual-band network architecture, which only considers licensed and mmWave bands [16], we first introduce a novel triple-band network structure by integrating the sub-6GHz unlicensed band. Compared to the traditional dual-band structure with only the mmWave band and licensed band, we can steer the traffic of mmWave and licensed bands to the additional sub-6GHz unlicensed band. In this way, we can relieve traffic congestion in the licensed band and compensate for the blockage sensitivity of mmWave communications, thus better ensuring quality-of-experience (QoE) for users. Based on the proposed architecture, centralized and distributed cooperative mechanisms are further developed to reduce the computational complexity of network operation.

The main contributions of this paper can be summarized as follows.

We propose a novel triple-band network structure which utilizes the licensed bands, sub-6GHz unlicensed bands, and mmWave bands to support various user types. Specifically, mmWave APs, small-cell base stations (SBSs), and WiFi APs are clustered into groups according to their geographic locations. Each cluster is separately controlled by a subordinate controller, which is also under the control of a superior controller. The superior controller can collect global network information and make comprehensive decisions in a centralized way, whereas each subordinate controller performs the corresponding resource management distributively.To further improve the network performance, we then propose a novel self-optimizing traffic steering strategy which can steer traffic to specific cells according to the dynamic network and traffic environments. The procedures of the proposed self-optimizing traffic steering mainly include three steps: information collection, mode selection, and decision making. The input parameters for mode selection in each subordinate controller include the feedback of the previous network reconfiguration response and the key performance indicators (KPIs) provided by the real-time environment and traffic information.Based on the proposed framework, self-optimizing traffic steering can be preformed flexibly to provide further benefits for both networks and users. To analyze the potential advantages, several use cases of the proposed self-optimizing traffic steering are discussed.The remainder of this article is organized as follows. In Section 2, we introduce the characteristics of 5G networks and devices before presenting a novel triple-band network structure. In Section 3, we propose a novel self-optimizing traffic steering strategy and investigate the corresponding use cases. Performance evaluation results are presented in Section 4 and the article is finally concluded in Section 5.

## 2. Overview of mmWave HetNets

In this section, we first introduce the characteristics of 5G networks and devices and then describe the system model. Afterwards, we propose a novel triple-band network structure for the mmWave HetNet.

### 2.1. Characteristics of 5G Networks and Devices

The mmWave HetNet will focus on enabling extremely high data rates and significant capacity enhancement, as well as creating a multi-function network that can connect new services and empower new user experience. To ensure the above functions, our design should support a wide variety of 5G devices, spectra, and networks, as described in the following.

#### 2.1.1. 5G Devices

Future 5G devices will enable various types of critical services with different requirements of data rate, latency, reliability, and security, including enhanced mobile broadband (eMBB), massive machine type of communication (mMTC), and ultra-reliable and low-latency communication (URLLC) scenarios. Therefore, different users with diverse QoS requirements should be associated with different types of access points.

#### 2.1.2. Spectra

The proposed mmWave HetNet consists of a wide range of available spectrum bands, from low bands (licensed bands) and middle bands (sub-6GHz unlicensed bands) to higher bands (mmWave bands). Due to various transmission power levels and channel characteristics, different spectra may have different network coverages, which brings stiff challenges to mobility management and network association.

#### 2.1.3. Networks

With the above available spectrum bands, there exist four kinds of networks, namely, the LTE network, LTE-licensed-assisted-access (LTE-LAA) (or LTE-U) network, WiFi network, and mmWave network. The main characteristics of these networks are summarized in Table 1 and elaborated upon in the following.

The LTE network is specifically designed to operate in the licensed spectrum under a centralized control of network protocols to prevent packet collision among subscribers. Therefore, LTE networks can provide stable transmission, reliable communication, and large signal coverage. However, the limitation of available licensed spectra is the major shortage of LTE networks. Hence, it can only support low data rate transmissions, e.g., voice communication, online conference, and location-based service.The WiFi network utilizes the sub-6GHz unlicensed band based on the carrier sense multiple access with collision avoidance (CSMA/CA) protocol that can reduce packet collision, such as the distributed contention-based IEEE 802.11-compliant standards. The main advantages of the WiFi network are its low cost, easy deployment, and quick/open user access. However, the quality-of-service (QoS) of WiFi users can be hardly guaranteed due to the distributive nature of WiFi protocols. Therefore, the WiFi network is mostly used for video, web browsing, and so on.The LTE-LAA (or LTE-U) network allows users to access both licensed and sub-6GHz unlicensed spectra under a unified LTE network infrastructure by carrier aggregation, which has already been defined in the 3GPP Release 13 standard. Combined with the benefits from both licensed and unlicensed spectra, the LTE-LAA network can provide better link performance, higher data rate and larger coverage than the legacy WiFi network. Meanwhile, it has a lower cost and is more bandwidth-rich as compared with the LTE network. However, the coverage of the LTE-LAA network is generally smaller than that of the licensed LTE network due to the large channel path loss in the sub-6GHz unlicensed band. Thus, LTE-LAA networks can support new applications with the data rate requirement of megabits per second.The mmWave network is mainly focused on the 28 GHz band and other bands ranging from 30 GHz to 300 GHz, which can offer bandwidths up to 850 MHz in 28 GHz and even greater in other bands combined with further gains via directional beamforming. Several standards have already been considered in various commercial wireless systems, such as IEEE 802.15.3c for indoor wireless personal area networks (WPAN) [17] and IEEE 802.11ad for wireless local area networks (WLAN) [18]. Although the mmWave network can provide sufficient bandwidths and gigabit-level communication services, such as high definition television (HDTV), ultra-high definition video (UHDV), and virtual reality, it also faces several challenges compared with other existing communication systems. The main challenges are summarized as follows: (1) **High propagation loss**: Due to the small wavelength of mmWave signals, their diffractions are quite weak. Therefore, the strength of received signal power might be very weak due to NLOS transmission. (2) **Sensitivity to blockage**: Certain materials such as concrete walls, furniture, human bodies, or even foliage will cause severe penetration loss and consequently degrade the signal quality. (3) **High hardware complexity**: Traditional analog-to-digital converters (ADCs) in microwave transceivers should be redesigned in order to fulfill the directional pencil-beam transmission of mmWave transceivers, leading to high hardware complexity.

### 2.2. MmWave HetNet Model

According to the features of 5G devices and networks described above, we consider an mmWave HetNet consisting of a macro base station (MBS) and several small-cell base stations (SBSs), WiFi APs, and mmWave APs, as shown in Figure 1. The whole network is within the coverage of the MBS. Due to the high complexity of such HetNets, we utilize a hierarchical structure to manage the network. In particular, several nearby SBSs, mmWave APs and WiFi APs are clustered into one group and are associated with a subordinate controller. Meanwhile, all subordinate controllers are connected with one superior controller. In our system, subordinate controllers are utilized to perform resource management and user association in the corresponding groups, whereas the superior controller collects information from all subordinate controllers and makes comprehensive decisions in a centralized way.

### 2.3. Triple-Band Network Structure

Within the proposed mmWave HetNet, we further introduce a novel triple-band network structure which utilizes the licensed bands, sub-6GHz unlicensed bands, and mmWave bands together to support various user types. The proposed network structure works in the following way. The superior controller first evaluates the data rate requirements of users according to the information provided by subordinate controllers. Thereafter, each user is associated with a proper network. Specifically, users with gigabit-level data rate requirements prefer the mmWave network; high-QoS-demanding users with megabit-level data rate requirements prefer the LTE-LAA network; low-QoS-demanding users with megabit-level or even lower data rate requirements prefer the WiFi network; and high-QoS-requirement users with lower data rate requirements prefer the LTE network. With the information of all networks, each user can select a suitable access point (mmWave AP, LTE SBS, or WiFi AP) according to the real-time system environment.

Thus far, we have introduced a novel hierarchical and triple-band network structure for the mmWave HetNet. The main benefits of our proposals are summarized as follows.

The proposed mmWave HetNet is very compelling, as the large available mmWave bandwidth enables extremely high data rate transmission and new service requirements.Centralized approaches generally have high computational complexity and large signalling overhead, whereas distributed approaches cannot effectively handle the fluctuation in system environments. Combining both centralized and distributed approaches, we can not only decrease the computational complexity but also guarantee the user performance.Compared with the widely used dual-band network structure, our proposal mainly has the following merits: (1) It can release the traffic burden in both licensed and mmWave networks. (2) It can improve the QoS of users with megabit-level data rate requirements by associating them with the more stable LTE-LAA networks. (3) It can ensure the performance of users with gigabit-level data rate requirements by associating them with the bandwidth-richer mmWave network.

To further improve the system performance, we should seek potential solutions to user association in the triple-band network structure. Traffic steering, which can be flexibly and easily utilized by steering different types of users to different networks, is one of the best approaches. In the next section, we propose a self-optimizing traffic steering strategy for the mmWave HetNet.

## 3. Self-Optimizing Traffic Steering

In this section, we first introduce the concept of traffic steering and then propose a self-optimizing traffic steering strategy for the mmWave HetNet.

### 3.1. The Concept of Traffic Steering

As described earlier, the 5G network has a more heterogeneous structure with various spectra. Different networks and spectra have different characteristics and can support different kinds of user devices. To balance the network load, the traditional traffic offloading technique offloads users from heavily loaded networks to lightly loaded networks. As an enhancement of traffic offloading, traffic steering can intelligently manipulate and manage the traffic across different networks and spectra to further optimize the network performance, improve the resource utilization, and deliver better services for users. The applications of traffic steering can be diverse and beneficial for many aspects, as summarized in the following:**Resource utilization:** Due to traffic dynamics, the number of users may exceed the capacity limitation in some cells. Hence, users with different data rate requirements can be steered to corresponding nearby cells with sufficient resources to improve resource utilization.**Interference mitigation:** With the changing environment, a large number of users in a single network may cause severe packet congestion and interference. Therefore, traffic steering can also be used to re-distribute users across networks to mitigate interference.**Energy saving:** By considering the variation in traffic requirements, some cells might be under-utilized. In such a scenario, we can switch off these cells to save energy and users in these cells can be consequently steered to nearby cells.

### 3.2. Self-Optimizing Traffic Steering for mmWave HetNet

Now, we introduce a novel self-optimizing traffic steering method to facilitate the implementation of mmWave HetNet. As illustrated in Figure 2, the procedures of the proposed self-optimizing traffic steering mainly include three steps: information collection, mode selection, and decision making. The input parameters for mode selection in each subordinate controller include the feedback of the previous network reconfiguration response and the key performance indicators (KPIs) provided by the real-time environment and traffic information. Here, KPIs can be some key network metrics, such as the success/failure rates of the handovers, dropped calls, blockages, and traffic types. After making an initial decision, the subordinate controllers transmit this information to the superior controller. By collecting the global information, the superior controller makes the final decisions.

From the network aspect, traffic can be steered among different networks based on their resource limitations, traffic loads, cell environments, and so on. Therefore, there are two types of traffic steering: inter-network traffic steering and intra-network traffic steering, as shown in Table 2.

**Inter-network traffic steering:** Traffic can be steered among different networks based on resource utilization, QoS requirement, and other factors. For example, when licensed resources are limited, users in the LTE network with megabit-level or lower data rate requirements can be steered to the sub-6GHz unlicensed spectrum. In particular, users with high QoS requirements should be steered to the LTE-LAA network, whereas those with low QoS requirements should be steered to the WiFi network. Moreover, when the traffic of the WiFi network is overcrowded, those users with high QoS requirements are steered to the LTE or LTE-LAA network. Furthermore, when unlicensed sub-6GHz resources for the LTE-LAA network are insufficient, some users are steered according to their QoS requirements. Specifically, users with extremely high QoS requirements can be steered to the LTE network, while those with relatively low QoS requirements can be steered to the WiFi network. Traffic steering from the mmWave network to other networks or from other networks to the mmWave network differs from the above discussions, as described in the following.

From the mmWave network to other networks:
₋As we have discussed before, the channel condition in the mmWave network is more fluctuant due to high propagation loss and susceptibility to blockage. Hence, users’ QoS requirements cannot be easily fulfilled, which impels some users to be steered to other networks.₋Since the coverage of each mmWave AP is generally small, high-mobility users usually cannot receive good service from the mmWave network. Therefore, those users should be steered to the LTE or LTE-LAA network that has a larger coverage.₋In some low-dense user scenarios, the deployment of mmWave APs is unnecessary since other networks are sufficient to support user requirements. In this situation, some mmWave APs can be switched off and users in those mmWave APs can be steered to nearby LTE, LTE-LAA, or WiFi networks.From other networks to the mmWave network:
₋For those applications with gigabit-level data rate requirements, steering them to the mmWave network is preferable.₋mmWave communications are suitable for users with relatively low mobility, and thus it is better to steer low-mobility users to the mmWave network to release the traffic burden in other networks.₋In the hot-spot area, interference is the main bottleneck for data rate enhancement. In this situation, some users can be steered to the mmWave network to alleviate interference.

**Intra-network traffic steering:** Within the same network, traffic can be further steered among different cells and frequency bands. In intra-network traffic steering, actions are triggered based on traffic loads, available spectrum resources, channel capacities, and other factors. Intra-network traffic steering among LTE or WiFi networks is very common and has been widely studied, whereas the traffic steering strategy among different mmWave networks is quite different, as elaborated upon in the following:As mentioned above, objects in daily environments cause blockage to mmWave channels. Hence, overcrowded users in an mmWave AP may lead to severe blockage and propagation loss. In this case, traffic steering is needed to realize load balance among mmWave APs.Due to the small coverage of mmWave APs, handover among mmWave APs occurs frequently for moving users. Therefore, traffic steering should be applied to select the most suitable mmWave AP according to users’ trajectories.Since traffic service varies significantly with time, geographic location, and social events, resource utilization might be inefficient in some mmWave APs. Therefore, intra-network traffic steering can be used to reconfigure users in lightly loaded mmWave APs and switch off those idle mmWave APs to save energy.

**Mathematical model:** Define binary variable $x_{k}^{ij}$ as the traffic steering indicator, where $x_{k}^{ij}=1$ indicates that the traffic of user *k* is steered to a nearby SBS/AP *j* in network *i* and $x_{k}^{ij}=0$ otherwise. Here, the superscript i={0,1,2} indicates the LTE, LTE-LAA, and mmWave networks, respectively. To ensure efficient and reliable network connectivity, we enable the traffic of user *k* to be steered to multiple networks. Furthermore, define $y_{ku}^{ij}$ as the binary indicator that decides whether the channel *u* at SBS/AP *j* in network *i* is assigned to user *k*, and define $p_{k}^{ij}$ as the corresponding transmit power allocated to user *k*. Let $R_{k}^{ij}$ denote the achievable data rate of user *k* at SBS/AP $j\in J^{i}$. Then, $R_{k}^{ij}$ can be modeled as
(1)$$R_{k}^{ij}=B^{i}\sum_{u\in U^{i}}y_{ku}^{ij}\log_{2}\left(1+\frac{p_{k}^{ij}g_{ku}^{ij}}{\sum_{k^{\prime }\neq k}y_{k^{\prime }u}^{ij}p_{k^{\prime }u}^{ij}g_{k^{\prime }u}^{ij}+\sigma^{2}}\right),$$
where $g_{ku}^{ij}$ indicates the channel gain from SBS/AP $j\in J^{i}$ to user *k* at channel *u* and $\sigma^{2}$ is the variance of additive white Gaussian noise (AWGN). Furthermore, $U^{i}$ and $B^{i}$ denote the channel set available for network *i* and the corresponding channel bandwidth, respectively.

Based on the above definitions, the proposed joint inter-network and intra-network traffic steering can be mathematically formulated as the following sum rate maximization problem:(2)$$P_{0}:\;\max_{x,y,p}\;\sum_{i\in I}\sum_{j\in J^{i}}\sum_{k\in K}x_{k}^{ij}R_{k}^{ij}$$
(3)$$s.t.\;\;\sum_{i\in I}\sum_{j\in J^{i}}x_{k}^{ij}\geq 1,\;\forall k\in K,$$
(4)$$\sum_{u\in U^{i}}y_{ku}^{ij}\geq x_{k}^{ij},\;\forall i\in I,\;\forall j\in J^{i},$$
(5)$$\sum_{k\in K}p_{k}^{ij}\leq P_{\max}^{i},\;\forall i\in I,\;\forall j\in J^{i},$$
(6)$$\sum_{i\in I}\sum_{j\in J^{i}}R_{k}^{ij}\geq R_{\min}^{k},\;\forall k\in K,$$
where I={0,1,2}, $J^{i}$ denotes the set of SBSs/APs in network *i*, and K indicates the user set. In problem $P_{0}$, constraint (Equation 3) signifies that the traffic of each user *k* should be steered to at least one network. Constraint (Equation 4) ensures that user *k* can access at least one channel if it is steered to SBS/AP $j\in J^{i}$, and constraints (Equation 5) and (Equation 6) guarantee the maximum transmit power of each SBS/AP $j\in J^{i}$ and the minimum data rate of each user k∈K.

The formulated sum rate maximization problem $P_{0}$ is a coupled mixed-integer non-linear programming problem (MINLP), which can be jointly solved by semi-distributed optimization algorithms with fractional programming, such as the semi-distributed alternating direction method of multipliers and block coordinate update method (ADMM-BCU) [19,20].

### 3.3. Use Cases

Based on the proposed framework, self-optimizing traffic steering can be performed flexibly to provide further benefits for both networks and users. To analyze the potential advantages, several use cases are further discussed in this subsection.

#### 3.3.1. Computational Complexity Decrement

Both intra-network and inter-network traffic steering involve multiple cells from either the same or different spectra. Furthermore, the resource occupation, user association, and channel status should be jointly considered based on the instantaneous user condition and channel information. Therefore, the optimization problem could be a very complicated mixed-integer non-linear program which is hard to solve. Moreover, the mmWave HetNet is usually ultra-densely deployed. Therefore, low-complexity traffic steering schemes are demanded for practical implementation. To tackle this problem, two efforts are made in this article. First, centralized and distributed approaches are combined together to solve the optimization problem. Specifically, SBSs, mmWave APs, and WiFi APs are clustered into groups according to their geographical locations, with each group connecting one subordinate controller. Meanwhile, a superior controller is introduced to control the operation of all subordinate controllers in order to guarantee the whole system performance. Secondly, we introduce the self-optimizing traffic steering strategy to automatically and intelligently make decisions according to a dynamic environment.

#### 3.3.2. Joint Intra-Network and Inter-Network Resource Allocation

The joint optimization of intra-network and inter-network resource allocation offers more opportunities to enhance resource utilization and thus can further improve system performance. According to the different spectrum types, there are mainly two kinds of inter-network resource allocation in the proposed HetNet. First, licensed resources can be allocated among LTE, LTE-LAA, and mmWave networks to guarantee the control signal transmission. Secondly, unlicensed sub-6GHz resources can be allocated to WiFi and LTE-LAA networks for resource sharing. On the other hand, intra-network resource allocation can be performed in the same network, where resources are reconfigured among different SBSs, WiFi APs, or mmWave APs, according to their specific requirements. Here, advanced optimization theories can be implemented to devise optimal resource allocation algorithms based on the traffic status and system environment.

#### 3.3.3. Energy Efficiency Improvement

Since mmWave APs are densely deployed in the HetNet, a large amount of energy is consumed. Therefore, flexible and agile mmWave APs’ sleeping management by the self-optimizing traffic steering strategy should be designed for energy saving. For example, lightly loaded mmWave APs can be switched off and the associated users can be steered to nearby APs according to their specific needs. For practical implementation, incentive schemes need to be designed to compensate the performance degradation of the steered users, whereby a certain balance can be achieved between mmWave APs and users. Such problems can be modeled as multi-player games and thus game theory tools can be applied to seek the optimized and equilibrated solutions among players. Meanwhile, machine learning, such as deep learning and statistical learning techniques, can also be utilized to solve the problem.

## 4. Performance Evaluation

In this section, numerical results are presented to show the performance advantage of the proposed self-optimizing traffic steering for the mmWave HetNet. We consider a HetNet with 20 mmWave APs and 5 SBSs located within the coverage of a macro base station. There are 20 LTE users, 20 LTE-LAA users, and 50 mmWave users randomly distributed in the network according to the Poisson Point Process (PPP), with data rate requirements of 500 Kbps, 20 Mbps, and 2 Gbps, respectively. The transmission power of all users is set to be 0.1 W. We assume that all users are equipped with multiple network interfaces so that they can be steered from one network to any other network. Three different kinds of spectra, i.e., the 2.6 GHz licensed band with a bandwidth of 1 MHz, the 5 GHz unlicensed band with a bandwidth of 20 MHz, and the 28 GHz mmWave band with a bandwidth of 850 MHz, are considered. The path loss model for mmWave communications follows the one in [21], while that for the LTE and LTE-LAA communications adopts a standard 3GPP model [22]. Furthermore, we use the Boolean scheme to reflect the impact of mmWave communication blockages, where the probabilities of NLOS and LOS links are set to be 0.2 and 0.8, respectively.

In our test, we mainly evaluate the enhancement of the system throughput and energy efficiency by the proposed triple-band network structure and traffic steering strategy. Specifically, we utilize the average required time slots to reflect the throughput performance. More throughput can be achieved if less time slots are required. Regarding the energy efficiency, we use the mmWave AP sleeping ratio as the performance metric since a larger sleeping ratio means higher energy efficiency.

In the test, we assume that different types of users randomly choose their associated access points initially and allow that a maximum of 20% users can be steered from the mmWave network to others. From the figure, we see that less mmWave bandwidth is required if traffic steering is applied. Moreover, compared with the intra-network traffic steering, applying both intra- and inter- networks can utilize more sub-6GHz time slots but less mmWave time slots to achieve the same user requirements. This result indicates that the proposed traffic steering strategy can achieve a higher resource utilization efficiency.

In Figure 3b, we further compare the average percentage of sleeping mmWave APs and the average required mmWave time slots with different switch off thresholds. In this test, the mmWave AP is switched off when the associated user number is below a given threshold. In addition, we compare the dual-band network structure which only consists of the licensed band and mmWave band. The results in this figure show that the average percentage of sleeping mmWave AP increases with the switch off threshold. Moreover, the proposed triple-band network structure achieves a higher mmWave AP sleeping ratio and a smaller mmWave time-slot occupancy than the traditional dual-band one. Since more mmWave APs can be switched off to realize comparable system performance, it can be verified that the proposed triple-band network achieves a more efficient wireless resource utilization than existing dual-band 5G mmWave heterogeneous networks.

## 5. Conclusions

This article aims to develop a novel network structure and a traffic steering strategy to facilitate the deployment of mmWave communications in 5G HetNet. We first summarize the characteristics of 5G devices, spectra, and networks. Then, we present a novel triple-band network structure which integrates licensed bands, sub-6GHz unlicensed bands, and mmWave bands to provide better QoS for users. To further enhance network performance, a novel self-optimizing traffic steering strategy is then developed which dynamically steers traffic to specific cells according to the network and traffic environments. Several use cases of the proposed self-optimizing traffic steering are also discussed. Our test results demonstrate that the proposed triple-band network structure and self-optimizing traffic steering strategy can achieve better throughput and energy efficiency performances than the conventional design.

## Figures and Tables

**Figure 1 sensors-22-07112-f001:**
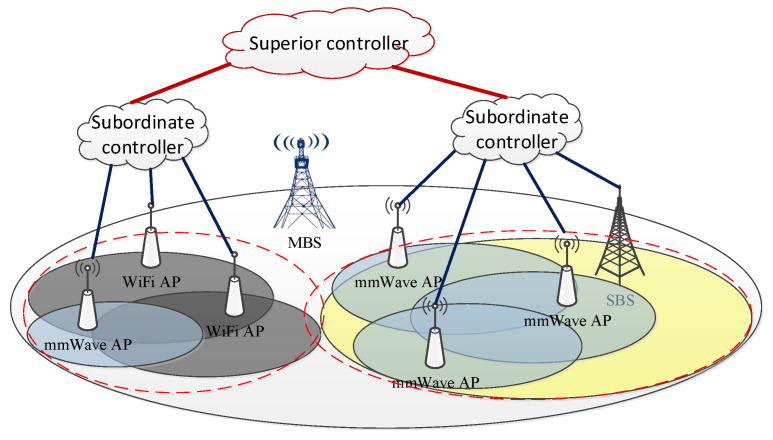
The proposed mmWave HetNet architecture.

**Figure 2 sensors-22-07112-f002:**
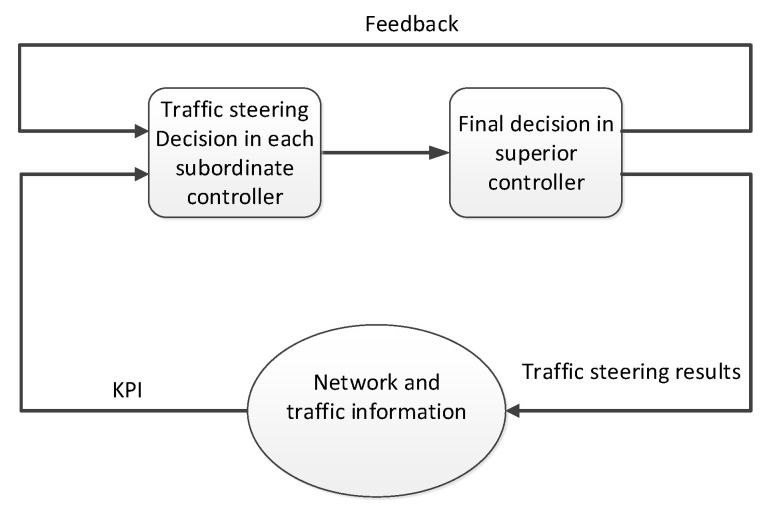
The general framework for self-optimizing traffic steering.

**Figure 3 sensors-22-07112-f003:**
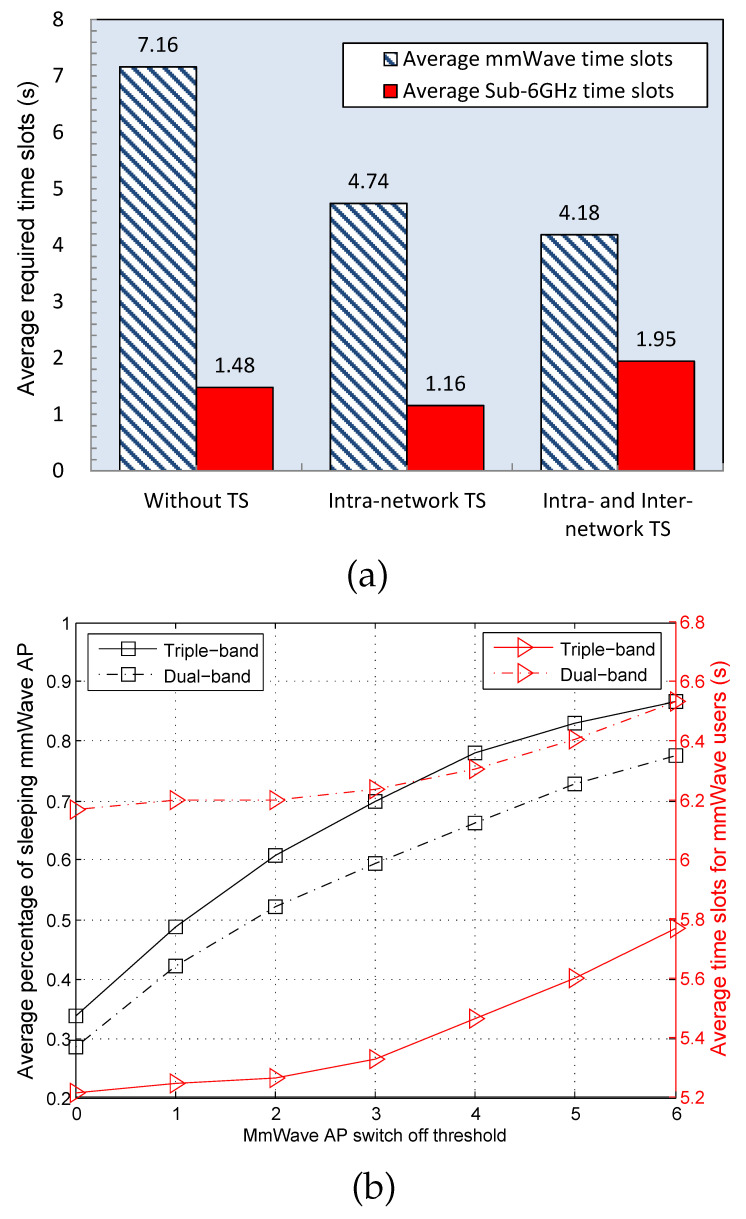
Simulation results: (**a**) Average required time slots in both mmWave and sub-6GHz unlicensed bands. (**b**) Average percentage of sleeping mmWave AP and the related time slots of mmWave users under different switch off thresholds.

**Table 1 sensors-22-07112-t001:** Comparison of different networks.

Candidate	Spectra	Merits	Demerits	Applications
LTE	Licensed	Stable transmission, reliable communication, and large signal coverage	Limited resources and low data rate	Voice communication, video conference, and location-based service
WiFi	Sub-6GHz unlicensed	Easy deployment, low cost, and quick access	Large traffic congestion, high dropping probability, and unstable QoS	Video and web browsing
LTE-LAA	Licensed + sub-6GHz unlicensed	Unified management, megabit data rate, larger coverage than WiFi, better QoS than WiFi, lower cost than LTE, and richer bandwidth than LTE	Larger path loss than LTE and harsher access than WiFi	New applications with hundreds of megabits per second data rate requirement
mmWave	Licensed + mmWave	Affluent resources, limited interference, millisecond delay, and gigabit data rate	Small coverage, high propagation loss, sensitivity to blockage, and complex hardware	New applications with multi-Gbps data rate requirement, such as HDTV, UHDV, and virtual reality

**Table 2 sensors-22-07112-t002:** Traffic steering modes in the proposed mmWave HetNet.

Types	Modes	Motivations
Inter-network traffic steering	LTE → LTE-LAA	LTE resource limitation.
	LTE → WiFi	Megabit-level or even lower data rate requirement.
	WiFi → LTE	Overcrowded traffic in WiFi network.
	WiFi → LTE-LAA	High QoS requirements.
	LTE-LAA → LTE	LTE-LAA resource limitation.
	LTE-LAA → WiFi	QoS requirement changes.
	mmWave → LTE-LAA	High propagation loss and easy-to-block channels.
	mmWave → LTE	High user mobility.
	mmWave → WiFi	Energy saving.
	LTE → mmWave	Gigabit-level data rate requirement.
	LTE-LAA → mmWave	Stable user movement.
	WiFi → mmWave	High user density.
Intra-network traffic steering	LTE-LAA → LTE-LAA	Overcrowded traffic in one LTE-LAA cell.
	WiFi → WiFi	The collision probability of one WiFi cell is too high.
	mmWave → mmWave	Overcrowded in one mmWave cell. Small mmWave coverage. Energy saving.

## Data Availability

Not applicable.

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
