# Peer review of "Self-Optimizing Traffic Steering for 5G mmWave Heterogeneous Networks"

_sensors, 2022, doi:10.3390/s22197112_

Round 1
Reviewer 1 Report
In reviewers opinion, proposed by the authors self-optimizing traffic steering strategy should be somehow formalized mathematically or algorithmically. Current work includes just verbal description, which lacks mathematical exactness. Also, in reviewers opinion, authors should more clearly provide contribution with quantitative estimates and comparison with known state of the art results in 5G mmWave heterogeneous networks.
Author Response
Please refer to the attachment for the specific response.

Reviewer 2 Report
Please see the attached file.

Author Response

(The authors gave the same response as above.)

Round 2
Reviewer 1 Report
I believe the manuscript has been sufficiently improved.
Reviewer 2 Report
The authors have addressed all my concerns, no further comments.